# Automatic Debate Evaluation with Argumentation Semantics and Natural Language Argument Graph Networks

**Ramon Ruiz-Dolz[1], Stella Heras[2], Ana García-Fornes[2]**

[1]Centre for Argument Technology, University of Dundee, Dundee DD1 4HN, United Kingdom
[2]VRAIN, Universitat Politècnica de València, 46022 València, Spain
`rruizdolz001@dundee.ac.uk`, {`stehebar,agarcia`}`@dsic.upv.es`

## Abstract

The lack of annotated data on professional argumentation and complete argumentative debates has led to the oversimplification and the inability of approaching more complex natural language processing tasks. Such is the case of the automatic evaluation of complete professional argumentative debates. In this paper, we propose an original hybrid method to automatically predict the winning stance in this kind of debates. For that purpose, we combine concepts from argumentation theory such as argumentation frameworks and semantics, with Transformer-based architectures and neural graph networks. Furthermore, we obtain promising results that lay the basis on an unexplored new instance of the automatic analysis of natural language arguments.

## 1 Introduction

The automatic evaluation of argumentative debates is a Natural Language Processing (NLP) task that can support judges in debate tournaments, analysts of political debates, and even help to understand the human reasoning used in social media (e.g., Twitter debates) where argumentation may be difficult to follow. This task belongs to the computational argumentation area of research, a broad, multidisciplinary area of research that has been evolving rapidly in the last years (Atkinson et al., 2017). Classically, computational argumentation research focused on formal abstract logic and computational (i.e., graph) representations of arguments and their relations. In this approach, the evaluation of arguments relied exclusively in logical and topological properties of the argument representations (Alfano et al., 2021). Furthermore, these techniques have been thoroughly studied and analysed, but from a theoretical and formal viewpoint considering specific cases and configurations instead of large, informal debates (Verheij, 2005; Caminada, 2006).

The significant advances in NLP have enabled the study of new less formal approaches to under-take argumentative analysis tasks (Lawrence and Reed, 2019; Goffredo et al., 2023). One of the most popular tasks that has gained a lot of popularity in the recent years is argument mining, a task aimed at finding argumentative elements in natural language inputs (i.e., argumentative discourse segmentation) (Jo et al., 2019), defining their argumentative purpose (i.e., argumentative component classification) (Bao et al., 2021), and detecting argumentative structures between these elements (i.e., argumentative relation identification) (Ruiz-Dolz et al., 2021a). Even though most of the NLP research applied to computational argumentation has been focused in argument mining, other NLP-based tasks have also been researched such as the generation of natural language arguments (Mitsuda et al., 2022), the assesment of the persuasiveness of natural language arguments (El Baff et al., 2020), and the automatic generation of argument summaries (Bar-Haim et al., 2020) among others. However, it is possible to observe an important lack of research aimed at the evaluation of complete argumentative debates approached with NLP-based algorithms. Furthermore, most of the existing research in this topic has been contextualised in online debate forums, considering only short text arguments and messages, and without a professional human evaluation (Hsiao et al., 2022).

In this paper, we propose a hybrid method for evaluating complete argumentative debates considering the lines of reasoning presented by professional debaters and taking into account the human evaluations provided by an impartial jury. Our method combines concepts from the classical computational argumentation theory (i.e., argumentation frameworks and semantics), with models and algorithms effectively used in other NLP tasks (i.e., Transformer-based sentence vector representations and graph networks). This way, we take a complete professional debate including all the argumentation and rebuttal phases as an input, and predict the win-

ning stance (i.e., in favour or against) for a given argumentative topic. For that purpose, we define an original computational modelling of complete professional natural language debates that allows us to improve the prediction of the winning stance of complete argumentative debates compared to more conventional approaches. We present a complete comparison of approaches relying exclusively on NLP algorithms and approaches relying exclusively on argumentation theory concepts. From our findings, we can observe that the hybrid approach proposed in this paper is the more adequate to tackle a complex NLP challenge such as predicting the winning stance in complete debates.

## 2 Related Work

Classically, the computational representation and assessment of arguments has been conducted through argumentation frameworks and argumentation semantics (Dung, 1995). However, this line of research has been focused on abstract argumentation and formal logic-based argumentative structures, and has not been properly extended to the *informal* natural language representation of human argumentation.

The automatic assessment of natural language arguments is a relatively new topic of research that has been addressed from different NLP viewpoints. Most of this research has been focused on performing an individual evaluation of arguments or argumentative lines of reasoning (Wachsmuth et al., 2017) instead of a global, *interactive* viewpoint where complete debates consisting of multiple, conflicting lines of reasoning are analysed. Typically, the automatic evaluation of natural language arguments has been carried out comparing the convincingness of pairs of arguments (Gleize et al., 2019); analysing user features such as interests or personality to predict argument persuasiveness (Al Khatib et al., 2020); and analysing natural language features of argumentative text to estimate its persuasive effect (El Baff et al., 2020). The use of graph-based approaches to evaluate individual argument structures has been recently explored in (Saveleva et al., 2021). In the same direction, (Marro et al., 2022) proposes a framework for evaluating three dimensions of arguments (i.e., cogency, rhetoric, and reasonableness) by producing natural language embeddings from individual argument structures (e.g., claim - premise). However, none of them considers the problem of argument evaluation

in an argumentative dialogue as in the case of the debates.

The global (i.e., debate) approach on the evaluation of natural language arguments was initially researched in (Potash and Rumshisky, 2017) where Recurrent Neural Networks were used to evaluate non-professional debates in a corpus of limited size and structure. Following this trend, in (Shirafuji et al., 2019), the authors propose a method based on the persuasiveness to predict the outcome of online debates using a support vector machine. Recently, in (Hsiao et al., 2022), the authors present an algorithm for predicting the outcome of non-professional debates of limited length and depth in online forums. Furthermore, in the previous work the considered argumentative structures are simple, and the proposed methods depend exclusively on natural language features. All these works have two main aspects in common: first, they are focused exclusively in online text-based debates, where information is easy to obtain, but very limited from an argumentative viewpoint; and second, the debates brought into consideration present short interactions and simpler arguments than the ones that can be found in a professional debate.

Interestingly, a recent trend in the proposed argument assessment algorithms from the more theoretical side of the computational argumentation area of research also consists of leveraging structural information of the graph to estimate the acceptability of arguments by using neural networks instead of classic solvers (Kuhlmann and Thimm, 2019; Malmqvist et al., 2020; Craandijk and Bex, 2021). However, in these cases arguments are treated as abstract entities without specific natural language being attributed to them, making their results more difficult to contextualise in a real situation with natural language arguments.

The previously reviewed work evidences that fundamental concepts from computational argumentation theory are typically overseen in the argument-related NLP literature, and the used corpora contain *debates* far removed from the concept of a professional debate. Thus, we propose a new method that combines the advantages of both areas of research: formal argumentation theory and NLP. This way, our proposal enables the analysis of complete professional argumentative debates in both, length and argumentative depth, a task that has not been addressed in the literature yet.

# 3   Data

In this paper, we approach the automatic prediction of the winning stance in complete natural language professional debates. For that purpose, we use the *VivesDebate*[1] corpus (Ruiz-Dolz et al., 2021b) to conduct all the experiments and the validation of our proposed method. This corpus contains the annotations of the complete lines of reasoning presented by the debaters in a debate tournament, inspired by the AIF (Chesnevar et al., 2006) standard, and the professional jury evaluations of the quality of argumentation presented in each debate. It is important to emphasise this aspect, as the average length of the debates we analysed in this paper is 4819 words (30-40 minutes of length), and large language models have problems when working with long sequences of natural language text (Beltagy et al., 2020)[2]. Previously published corpora for the analysis of natural language argumentation always tended to simplify the annotated argumentative reasoning, by only considering individual arguments, pairs of arguments, or considering a small set of arguments, instead of deeper and complete lines of argumentative reasoning. For example, in argument mining (e.g., *US2016* (Visser et al., 2020) and *QT30* (Hautli-Janisz et al., 2022)), argument assessment (e.g., *IBM-EviConv* (Gleize et al., 2019)), or natural language argument generation/summarisation (e.g., *GPR-KB (Orbach et al., 2019), DebateSum* , (Roush and Balaji, 2020)). Furthermore, online debates with their crowd-sourced evaluations were compiled in (Durmus and Cardie, 2019), but argumentation was produced in short written paragraphs, and evaluations were based on anonymous votes from the community that did not require any justification. Therefore, the *VivesDebate* corpus is the only identified publicly available corpus that enables the study of the automatic evaluation of natural language professional debates in their complete form.

The *VivesDebate* corpus contains 29 complete argumentative debates (139,756 words) from a university debate tournament in Catalan. Each debate is annotated entirely without partitions, and capturing the complete lines of reasoning presented by the debaters. The natural language text is segmented into Argumentative Discourse Units (7,810 ADUs) (Peldszus and Stede, 2013). Each ADU

contains its own text, its stance (i.e., in favour or against the topic of the debate), the phase of the debate where it has been uttered (i.e., introduction, argumentation, and conclusion), and a set of argumentative relations (i.e., inference, conflict, and rephrase) that make possible to capture argumentative structures, the sequentiality in the debate, and the existing major lines of reasoning. Additionally, each debate has the scores of the jury that indicate which team has proposed a more solid and stronger argumentative reasoning. An in-depth analysis of the corpus structure and statistics can be found in (Ruiz-Dolz et al., 2021b).

# 4   Method

The human evaluation of argumentative debates is a complex task that involves many different aspects such as the thesis solidity, the argumentation quality, and other linguistic aspects of the debate (e.g., oral fluency, grammatical correctness, etc.). It is possible to observe that both, the logic of argumentation and the linguistic properties play a major role in the evaluation of argumentative debates. Therefore, the method proposed in this paper is designed to capture both aspects of argumentation by combining concepts from computational argumentation theory and NLP. Our method is divided into two different phases: first, (i) determining the acceptability of arguments (i.e., their logical validity) in a debate based on their logical structures and relations; and second, (ii) scoring the resulting acceptable arguments by analysing aspects of their underlying natural language features to determine the winner of a debate. Figure 1 presents an scheme with the most important phases and elements of the proposed method.

Before describing both phases of our method, it is important to contextualise our proposal within the area of computational argumentation research. We assume that the whole argument analysis of natural language text has already been carried out: the argumentative discourse has been segmented, the argument components have been classified, and argument relations have been identified among the segmented argumentative text spans (see (Lenz et al., 2020)). Thus, a graph structure can be defined from a given natural language argumentative input. As depicted in Figure 1, the *Argument Analysis* containing the text of the arguments (i.e., node content), their stance (i.e., node colour), inference relations (i.e., green edges), conflict relations

---

[1]Available online in: https://doi.org/10.5281/zenodo.6531487
[2]The Longformer supports sequences of up to 4096 tokens.

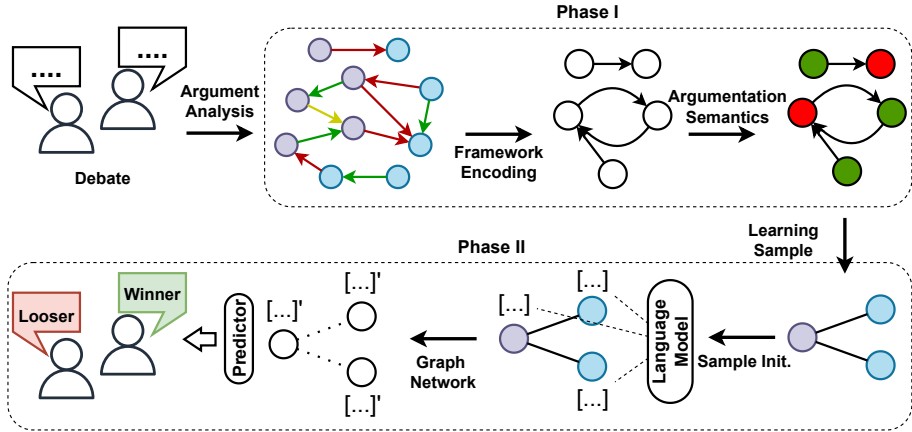

Figure 1: Structural scheme of the proposed automatic debate evaluation method.

(i.e., red edges), and rephrase relations (i.e., yellow edges) among arguments can be a valid starting point to the proposed method.

(Lenz et al., 2019)

### 4.1 Phase I: Argument Acceptability

The first phase of the proposed method relies on concepts from computational argumentation theory. This phase can be understood as a pre-processing step from the NLP research viewpoint. Thus, the main goal of *Phase I* is to analyse the argumentative information contained in the argument graph, and to computationally encode this information focusing on the most relevant aspects for natural language argumentation (see Figure 1, *Framework Encoding* and *Argumentation Semantics*).

For that purpose, it is necessary to introduce the concept of an abstract argumentation framework and argumentation semantics. Originally proposed by Dung in (Dung, 1995), an argumentation framework is a graph-based representation of *abstract* (i.e., non-structured) arguments and their attack relations:

**Definition 1 (Argumentation Framework)** *An Argumentation Framework (AF) is a tuple $AF = < A, R >$ where: $A$ is a finite set of arguments, and $R$ is the attack relation on $A$ such as $A \times A \rightarrow R$.*

Furthermore, argumentation semantics were proposed along with the AFs as a set of logical *rules* to determine the acceptability of an *abstract* argument or a set of arguments. In this paper, following one of the most popular notations in argumentation theory, we will refer to these sets as acceptable extensions. These semantics rely on two essential set properties: conflict-freeness and admissibility.

Thus, we can consider that a set of arguments is conflict free if there are not any attacks between arguments belonging to the same set:

**Definition 2 (Conflict-free)** *Let $AF = < A, R >$ be an argumentation framework and $Args \subseteq A$. The set of arguments $Args$ is conflict-free iff $\neg \exists \alpha_i, \alpha_j \in Args : (\alpha_i, \alpha_j) \in R$.*

We can also consider that a set of arguments is admissible if, in addition of being conflict-free, it is able to defend itself from external attacks:

**Definition 3 (Admissible)** *Let $AF = < A, R >$ be an argumentation framework and $Args \subseteq A$. The set of arguments $Args$ is admissible iff $Args$ is conflict-free, and $\forall \alpha_i \in Args, \neg \exists \alpha_k \in A : (\alpha_k, \alpha_i) \in R$ and $(\alpha_i, \alpha_k) \notin R$, or $\neg \exists \alpha_j \in Args : (\alpha_j, \alpha_k) \in R$*

In this paper, we compare the behaviour of these two properties through the use of Naïve (conflict-free) and Preferred (admissible) semantics to compute all the acceptable extensions of arguments from the AF representations of debates. Naïve semantics are defined as maximal (w.r.t. set inclusion) conflict-free sets of arguments in a given AF. Similarly, Preferred semantics are defined as maximal (w.r.t. set inclusion) admissible sets of arguments in a given AF.

At this point, it is important to remark that the criteria of selecting both semantics for our method is oriented by the principle of maximality. Since acceptable extensions will be used as samples to train the natural language model in the subsequent phase of the proposed method, we selected these semantics that allow us to obtain the highest number of extensions, but keeping the most of the natural language information and maximising differences

among the extensions (i.e., not accepting the subsets of a given maximal extension, which would result in data redundancy and hamper the learning process of the model).

---

**Algorithm 1** Argumentation Framework Encoding.

1: **function** GRAPHTOAF($ArgumentGraph$)
2:      $AG \leftarrow ArgumentGraph$
3:      $r \leftarrow AG.edges('conflict')$
4:      $AG.removeEdges(r)$
5:      $cc \leftarrow AG.connected\_components()$
6:      $AF \leftarrow NewGraph()$
7:      **for** $subgraph \in cc$ **do**
8:          $arg \leftarrow \{\}$
9:          **for** $node \in subgraph$ **do**
10:            $arg.append(node.Data())$
11:          **end for**
12:          $AF.addNode(arg)$
13:      **end for**
14:      $AF.addEdges(r)$
15:      **return** $AF$
16: **end function**

---

Therefore, we encode the argument graphs, resulting from a natural language analysis of the debate, as abstract AFs using the proposed Algorithm 1. ADUs that follow the same line of reasoning (i.e., related with inference or rephrase) are grouped into *abstract* arguments, and the existing conflicts between ADUs are represented with the attack relation of the AF. Then, both Naïve and Preferred semantics are computed on the AF representation of the debate. This leads to a finite set of extensions, each one of them consisting of a set of acceptable arguments under the logic *rules* of computational argumentation theory. These extensions will be used as learning samples for training and evaluating the natural language model in the subsequent phase of the proposed method.

### 4.2 Phase II: Debate Outcome Estimation

The second phase of the method focuses on analysing the natural language arguments contained in the acceptable extensions, and determining the winner of a given debate. For that purpose, we use the Graph Network (GN) architecture combined with Transformer-based sentence embeddings generated from the natural language arguments contained in the acceptable extensions. A GN is a machine learning algorithm aimed at learning computational representations for graph-based data structures (Battaglia et al., 2018). Therefore,

a GN receives a graph as an input containing initialised node features (i.e., $v_1, \ldots, v_i \in V$), edge data (i.e., $(e_1, r_1, s_1), \ldots, (e_k, r_k, s_k) \in E$, where $e$ are the edge features, $r$ is the receiver node, and $s$ is the sender node), and global features (i.e., $u$); and updates them according to three learnt update $\phi$ and three static aggregation $\rho$ functions:

$$
\begin{aligned}
e'_k &= \phi^e(e_k, v_{r_k}, v_{s_k}, u) & \bar{e}'_i &= \rho^{e \to v}(E'_i) \\
v'_i &= \phi^v(\bar{e}'_i, v_i, u) & \bar{v}' &= \rho^{e \to u}(E') \quad (1) \\
u' &= \phi^u(\bar{e}', \bar{v}', u) & \bar{v}' &= \rho^{v \to u}(V')
\end{aligned}
$$

This way, $\phi^e$ computes an edge-wise update of edge features, $\phi^v$ updates the features of the nodes, and $\phi^u$ is computed at the end, updating global graph features. Finally, $\rho$ functions must be commutative, and calculate aggregated features, which are used in the subsequent update functions.

Thus, the first step in *Phase II* is to build the learning samples from the previously computed extensions of AFs (see Figure 1, *Learning Sample*). An extension is a set of logically acceptable arguments under the principles of conflict-freeness and/or admissibility. However, there are no explicit relations between the acceptable arguments, since AF representations only consider attacks between arguments, and the conflict-free principle states that there must be no attacks between arguments belonging to the same extension. Thus, in order to structure the data and make it useful for learning linguistic features for the debate evaluation task, we generate a complete bipartite graph from each extension. The two disjoint sets of arguments are determined by their stance (i.e., one set consisting of all the acceptable arguments in favour, and the other against), since argumentation semantics allow to define sets of logically acceptable arguments but do not guarantee that they will have the same claim or a similar stance.

The second step consists on initialising all the required features of the learning samples for the GN architecture (see Figure 1, *Sample Init.*). Thus, we define which features will encode edge, node, and global information of the previously processed bipartite graph samples. Edges do not contain any relevant natural language information, so we initialise edge features identically (similar to previous research (Craandijk and Bex, 2021)), so that node influence can be stronger when learning edge update functions. Nodes, however, are a pivotal aspect of this second phase since they contain all

the natural language data. Node features are initialised from sentence embedding representations of the natural language ADUs contained in each node. Thus, we propose the use of a pre-trained language model to generate dense vector representations of these ADUs, and initialise the vector features for learning the task. Finally, the global features of our learning samples encode the probability distribution of winning/losing a given debate (represented as acceptable extension-based bipartite graphs), and are a binary label that indicates the winning stance (i.e., 0 for the team in favour and 1 for the team against).

The final step in the *Phase II* of our proposed method is focused on learning the automatic evaluation of argumentative debates (see Figure 1, *Graph Network*). In a classical debate, there are always two teams/stances: in favour and against some specific claim. In this paper, we approach the debate evaluation as a binary classification task. Therefore, at the end of the proposed method, we model the classification problem as follows:

$$\hat{c} = \underset{c \in C}{\arg\max} \, P(c|G) \qquad (2)$$

where $C = [\text{“}F\text{”}, \text{“}A\text{”}]$, depending on the winner of each debate (i.e., in "$F$"avour or "$A$"gainst). And $G$ is a complete bipartite graph generated from the acceptable extensions of the AF pre-processing described in the *Phase I* of our method. We approach this probabilistic modelling with three Multi Layer Perceptrons (MLP) consisting of two layers of 128 hidden units for each of the $\phi$ update functions. Since the debate evaluation is an instance of the graph prediction task, it is important to point out that the architecture of the two MLP approaching $\phi^e$ and $\phi^v$ are equivalent, and their parameters are learnt from the backpropagation of the MLP architecture for $\phi^u$. Finally, the proposed GN model has a 2-unit linear layer (for binary classification) and a *softmax* function (for modelling the probability distribution) on its top.

## 5 Experimental Analysis

### 5.1 Experimental Setup

All the experiments and results reported in this paper have been implemented using *Python 3* and run under the following setup. The initial corpus pre-processing and data structuring (i.e., *Phase I*) has been carried out using *Pandas* (McKinney et al., 2010) together with *NetworkX* (Hagberg

et al., 2008) libraries. Argumentation semantics have been implemented considering the *NetworkX*-based AF graph structures. Regarding *Phase II*, the language model and the dense sentence vector embeddings have been implemented through the *Sentence Transformers* library (Reimers and Gurevych, 2019). We used a pre-trained XLM-RoBERTa architecture (Reimers and Gurevych, 2020) able to encode multilingual natural language inputs into a 768 dimensional dense vector space (i.e., word embedding size). Finally, the *Jraph*[3] library has been used for the implementation of the graph network architecture, for learning its update functions (i.e., Equation 1), and for the probabilistic modelling defined in Equation 2. We used an Intel Core i7-9700k computer with an NVIDIA RTX 3090 GPU and 32GB of RAM to run all our experiments. The code implementation of the proposed method and the subsequent experiments is publicly available in `https://github.com/raruidol/ArgumentEvaluation`.

It is also important to completely define the notion behind a learning sample in the experimental setup, and how the data pipeline manages all these samples and structures them for training/evaluation. In our proposal, we defined a learning sample as an acceptable extension of a given debate. Thus, different debates may produce a different number of learning samples depending on the argumentation semantics and/or the argumentation framework topology. This way, learning samples can be managed from a debate-wise or an extension-wise viewpoint. Even though we used the learning samples individually (i.e., extension-wise) for the training of the proposed models, we will always consider debate-wise partitions of our data in our experimental setup. This decision has been made because it would be unfair to consider learning samples belonging to the same debate in both our train and test data partitions. The reported results could be misleading, and would not properly reflect the strengths and weaknesses of our method. Therefore, we used 29 complete debates in our experiments (18 in favour, 11 against). To create the train-test data splits, we assigned 23 debates (80%) to the train split and 6 debates (20%) to the test split.

To evaluate and validate our proposal, we have calculated the values of the performance scores averaged after 3 sequential runs with a different

---

[3] `https://jraph.readthedocs.io/`

(random) initialisation. As for the performance scores, we have calculated the precision, the recall, and the weighted average F1 score in all of our experiments. We decided to report the weighted average of the F1 score due to the existing variability of the class distributions on the learning samples generated from different debates included in the test split from one run to another.

## 5.2 Baselines

We defined five baselines to compare the performance of our proposed method and validate our contribution. The selected baselines have been defined to provide a better understanding of the benefits of using transversal approaches to a problem such as the prediction of the winning stance in a complete and long argumentative debate, where complex natural language and argumentative relations are present.

For that purpose, we have used a random baseline (RB) that assigned randomly a class to each debate as our most basic baseline. Moreover, we have considered two baselines that rely exclusively on concepts from computational argumentation theory (i.e., semantics) to determine the winner of a debate. These are the Naïve argumentation theory baseline (Naïve-ATB) and the Preferred argumentation theory baseline (Preferred-ATB), which compute a diferent set of acceptable extensions based on the different admissibility principles, and then calculate the majoritarian set of arguments grouped by stance. Therefore, the winning team is decided by having more acceptable arguments in the extensions produced by the argumentation semantics. To compare argumentation theory exclusive approaches with algorithms relying exclusively in NLP techniques, we have defined two more baselines. The first one is the Longformer (Beltagy et al., 2020) architecture, which we have fine-tuned to predict the winner of a debate directly from the text transcripts of the debates. We decided to use this model since it represents one of the best options to deal with longer texts in the state-of-the-art. We fine-tuned the model[4] for 3 epochs on our training data with a learning rate of 5e-5. As for the second natural language baseline, we have used the Graph Network Baseline (GNB), in which we ignored the *Phase I* of our proposed method and we trained the graph network directly on the argument

---
[4]https://huggingface.co/allenai/longformer-base-4096

graphs resulting from the argumentative analysis.

## 5.3 Our method

Apart from the baselines, we have experimented with two variants of our proposed method: the Naïve-GN and the Preferred-GN. In the Naïve-GN, we applied the Naïve semantics during the *Phase I*, and then trained the graph network using the learning samples generated based on the principle of conflict-freedom. Conversely, in the Preferred-GN approach, we calculated the Preferred extensions during *Phase I* and trained the graph network with the learning samples resulting from applying the principle of admissibility to our argumentation graphs. In the end, we obtained a total number of 471 Naïve and 32 Preferred extensions from the 29 debates. The 471 Naïve extensions were distributed as follows: 203 learning samples belonging to class 0 (i.e., in favour team wins in the 43.1% of the cases), and 268 samples belonging to class 1 (i.e., against team wins in the 56.9% of the cases). On the other hand, the 32 Preferred extensions were distributed as follows: 19 learning samples belonging to class 0, and 13 samples belonging to class 1.

From the variations in data distributions in both approaches, it is possible to observe how the admissibility principle is much more strict from a logical perspective than the conflict-free principle, and has a significant repercussion on the number of acceptable extensions (i.e., learning samples) produced. Therefore, more learning samples are produced by the Naïve semantics which can be leveraged by the *Phase II* of the proposed method.

## 5.4 Results

As we can observe in Table 1, the best results in all the evaluation metrics were obtained by the Naïve-GN approach. The Preferred-GN was the second best in terms of precision and recall, but performed similar to the argumentation theory baselines in terms of F1. Argumentation theory baselines, however, performed similar to the random baseline in all the metrics. This observation is telling us that relying exclusively on formal logic aspects in informal environments such as natural language debates, must not be enough to approach linguistically complex tasks such as the automatic evaluation of natural language argumentation. Finally, we have also been able to observe that both baselines relying exclusively on NLP algorithms and techniques performed worse than the random baseline. The main

| Experiment | Evaluation Metrics | | |
|---|---|---|---|
| Model | Precision | Recall | Weighted-F1 |
| RB | 0.59 | 0.55 | 0.55 |
| Naïve-ATB | 0.58 | 0.46 | 0.48 |
| Preferred-ATB | 0.51 | 0.55 | 0.53 |
| Longformer | 0.50 | 0.25 | 0.33 |
| GNB | 0.29 | 0.44 | 0.33 |
| **Naïve-GN** | **0.64** | **0.65** | **0.64** |
| Preferred-GN | 0.62 | 0.61 | 0.52 |

Table 1: Precision, recall, and weighted-F1 results of the automatic debate evaluation task. The reported results have been averaged from 3 randomly initialised sequential runs.

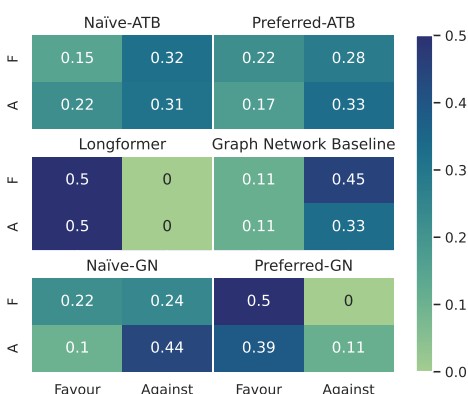

Figure 2: Aggregated confusion matrices.

cause of this problem can be probably attributed to the lack of data in our domain. State-of-the-art NLP algorithms rely on large amounts of data, which we do not have when it comes to analysing and evaluating complete professional debates in natural language.

We have also detected some interesting findings when looking specifically at each group of experimental baselines. Regarding the ATBs, it is possible to observe how, after 3 runs, the Preferred-ATB is consistently providing better results than the Naïve-ATB. This finding makes a lot of sense, since (despite their bad general performance) relying on the principle of admissibility is more informative from the argumentative viewpoint than doing it on the principle of conflict-freedom. A similar behaviour can be found within the NLP baselines. We can observe that the GNB has a significantly better recall than the Longformer, meaning that the GNB is doing a better generalisation than the Longformer. This behaviour can be attributed to the fact that, apart from the limitations in the size of the corpus for relying exclusively on state-of-the-art NLP techniques, learning representations from graph-structured data is a better idea than just using the whole text (i.e., debate transcripts) as the unique input for the models.

### 5.5 Error Analysis

Apart from looking at the performance scores, we have also analysed the behaviour of the proposed method and the baselines from the error perspective. Figure 2 depicts the aggregated confusion matrices of the three runs reported in Table 1. To

represent the error distributions, we have calculated the percentages after 3 runs of debates classified as in Favour (F) or Against (A) winning stances. We can observe how the ATBs present an almost uniform distribution of the error, similar to what can be expected of a random baseline. This goes in line with the reported results in the previous section. Conversely, the Longformer assigned to every debate the F winning stance (i.e., the majority class) giving more emphasis on the needs for larger corpora to make use of LLMs. Finally, we can observe the impact of the differences between the use of Naïve and Preferred semantics in our methods on the error distributions. With the Naïve-GN, the model managed to correctly identify almost all the debates where A was the winning stance, but had some problems when doing it in Favour. On the other hand, the Preferred-GN did the opposite, all the debates where F was the winning stance were correctly classified, but in the case of A debates, the model miss-classified most of them.

## 6 Discussion

In this paper, we have defined an original hybrid method to approach the winning stance prediction of complete natural language argumentative debates. For that purpose, we present a new instance of the debate assessment task, where argumentative debates and their underlying lines of reasoning are considered in a comprehensive, undivided manner. The proposed method combines aspects from formal logic and computational argumentation theory, with NLP and Deep Learning. From the observed results, several conclusions can be drawn. First, it has been possible to determine that our method performed better than approaching independently

the debate evaluation task from either the argumentation theory or the NLP viewpoints. Furthermore, we have observed in our experiments that conflict-free semantics produce a higher number of acceptable extensions from each AF compared to the admissibility-based semantics. This helped to improve the learning process of the task in a similar way to that achieved by data augmentation techniques. Thus, a better probabilistic models of natural language distributions that are not too constrained to formal logic and graph topology can be learnt by our model.

This paper represents a solid starting point of research in the evaluation of complete natural language professional debates. As future work, we are interested in considering finer-grained features for the evaluation of argumentation such as thesis solidity, argumentation quality, and adaptability. We also plan to extend our method with acoustic features, considering aspects such as the intonation or the fluency.

## Limitations

The present paper describes a new method for the automatic evaluation of complete argumentative debates. However, several limitations are described along the paper. Even though the proposed method can be generalised to any argumentative domain, the reported results are constrained to the domain of the corpus used in the experiments to validate our proposal. As discussed in Section 3, it has not been possible to identify any other corpus suitable for approaching the task as presented in this work. Thus, we are not able to evaluate our approach when considering different argumentation topics or domains. Furthermore, we used different baselines to validate the improvements achieved by our proposed method, but it was not possible to use previous research as reference. Finally, since we used real debates, the experimental configuration was also highly dependent on the specific debate structures. Extending the corpus may help to provide more solid results and explore the performance of our proposal when generalising to multiple topics and domains.

## Acknowledgements

This work has been developed thanks to the funding of the following projects: Grant PID2021-123673OB-C31 funded by MCIN/AEI/10.13039/501100011033, Grant TED2021-131295B-C32 funded by AEI/10.13039/501100011033/ European Union NextGenerationEU/PRTR, Spanish Government project PID2020-113416RB-I00, and the 'AI for Citizen Intelligence Coaching against Disinformation (TITAN)' project, funded by the EU Horizon 2020 research and innovation programme under grant agreement 101070658, and by UK Research and innovation under the UK governments Horizon funding guarantee grant numbers 10040483 and 10055990.

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
