# OpenReview forum: "Automatic Debate Evaluation with Argumentation Semantics and Natural Language Argument Graph Networks"
_EMNLP/2023/Conference — EMNLP 2023 Main_

### Official Review · Reviewer_6Vnd · 2023-08-05

**Typos Grammar Style And Presentation Improvements:** Spelling error on line 654 - miss-cla…
**Soundness:** 3

**Excitement:**

3: Ambivalent: It has merits (e.g., it reports state-of-the-art results, the idea is nice), but there are key weaknesses (e.g., it describes incremental work), and it can significantly benefit from another round of revision. However, I won't object to accepting it if my co-reviewers champion it.

**Missing References:**

In section 5.2 the author/s discuss Argument Theory Baselines and subsequently utilize them in table 1. I was not able to find a citation for this and the author/s do not go into details of what these baselines entail. Either a citation or some details on the baselines used would be appreciated here.

**Paper Topic And Main Contributions:**

This paper proposes a new method to evaluate natural language debates based on a hybrid approach of combining argumentation theory and NLP techniques. The approach taken seems to be an interesting one, with the potential of leveraging the improvements made in both fields to perform the challenging task of debate outcome prediction. The model used is, to my knowledge, an original one and comparisons are made to reasonable baselines although some details are missing. The two phase approach taken to argument scoring is another good contribution.

**Questions For The Authors:**

A. How is the removal of arguments having an attack relation on them justified?
B. Question on extension of work - often debates are not neatly categorized into phases of speech and rebuttal - how does this architecture handle that?

**Reasons To Accept:**

1. Interesting two phase approach to argument quality detection.
2. Innovative model proposed to handle debate outcome prediction,

**Reasons To Reject:**

There are some decisions taken by the author/s that I would like some clarifications on, as to my understanding, they impact the outcome of the paper in a major way.
1. The removal of arguments that have an attack relation on them prior to them being scored strikes me as being quite extreme. There may be a good argument that has a substandard attack on it. Removal of that argument from the set of admissible arguments is likely to make the model non representative, as human adjudicators may still credit the argument. It may also be the case that an argument is too bad to be attacked. Most debates have a defined time for speakers to rebut their opposition and often the arguments that are focused on are the ones most fundamental to the case that the opposition is running. It makes little sense at that point to spend valuable time on rebutting throwaway statements. In the case of the method proposed by the authors, a good argument that is rebutted badly would be inadmissible and a bad argument that is deemed not worth rebutting would be admissible. This may cause issues in the subsequent steps.
2. I am unclear on how arguments impact the probability of winning the debate. While section 4.2 is titled argument scoring, I am not clear where that process is actually happening.

**Reproducibility:**

2: Would be hard pressed to reproduce the results. The contribution depends on data that are simply not available outside the author's institution or consortium; not enough details are provided.

**Reviewer Confidence:**

3: Pretty sure, but there's a chance I missed something. Although I have a good feel for this area in general, I did not carefully check the paper's details, e.g., the math, experimental design, or novelty.

---

> ### Author Rebuttal · Authors · 2023-08-24
>
> Dear reviewer 6Vnd,
>
> First of all we would like to thank you for your insightful feedback and constructive comments. In our response, we would like to address the two mentioned reasons to reject and the questions with hopes of improving your impression in our work.
>
> Regarding the first concern and reason to reject, we would like to provide further clarifications. First of all, we have evaluated two different sets of argumentation semantics: Naive and Preferred. The former are based on the concept of conflict-freeness, meaning that all the different sets of arguments that do not contain conflicting arguments are acceptable. The later are based on the concept of admissibility, meaning that in addition to being conflict free, the set of acceptable arguments must be able to defend themselves from the attacking arguments. Let us illustrate these two semantics with a very basic example, let us assume argument A attacking argument B. Considering the Naive semantics, we will have two sets of acceptable arguments {A} and {B}. Conversely, considering the Preferred semantics we will only have one set of acceptable arguments: {A}. Therefore, our proposed method does not remove arguments that have an attack relation on them, only the implementation with Preferred semantics might do that. Furthermore, the best performing implementation of our proposed method is the one based on the Naive semantics, and among other possible reasons this might be one of them to make it more successful (as discussed in the paper).
>
> On the other hand, the second major concern is the specific impact of atomic arguments to the probability of winning/loosing a debate. In our method, we use the Graph Network architecture to automatically score the whole debate rather than scoring individual arguments and aggregating these scores to predict the winner/looser. Therefore, the GN takes into consideration the natural language included in the arguments and the conflicting relations to learn the parameters and estimate the outcome of a debate. However, we agree that the name of the section may be misleading for the reader and we will modify it for the camera ready version of the paper to: 4.2 Debate Outcome Estimation.
>
> Finally, we would like to answer the questions. Regarding the question A, we provided a clear justification of it in the first paragraph of our response. Regarding the question B, our proposed method is robust to variations in the structure of the debates, since our algorithms are not using explicit information of the debate structure as their input. We just need an argument graph (e.g., the outcome of an argument mining pipeline) representing the debate as an input to run our proposed method.
>
> We would also like to mention that we will add further references to make more clear the argumentation theory baselines and improve the understandability of our work. Furthermore, regarding the reproducibility score, we will release the code upon acceptance making it easy to reproduce our experiments.
>
> We hope that with this thorough response we will be able to improve your Excitement about our work and improve your perception about its Soundness.
>
> Thank you very much again for your time and dedication,
>
> The authors.

---

### Official Review · Reviewer_5zCQ · 2023-08-05

**Soundness:** 4

**Excitement:**

4: Strong: This paper deepens the understanding of some phenomenon or lowers the barriers to an existing research direction.

**Paper Topic And Main Contributions:**

The article presents an original method for evaluating the winning stance in complete debates, under real-world circumstances involving professional debaters. The proposed method is a hybrid approach that combines argumentation frameworks and semantics on one side, and Graph-Network and Transformer-based architectures on the other.

The argumentation frameworks module categorizes arguments as either Conflict-free or Admissible. Subsequently, the arguments are processed by a hybrid model that integrates a Graph-Network architecture with embeddings generated by a Transformer model. The winner is estimated as the participant with the highest count of acceptable arguments.

**Questions For The Authors:**

- You wrote that "both baselines relying exclusively on NLP algorithms and techniques performed worse than the random baseline. The main cause of this problem can be probably attributed to the lack of data in our domain". But did you consider testing simpler probabilistic models for NLP, such as a Bayesian classifier, which performs well with small datasets?

- Did you consider comparing the results of your method with those of a generative LLM, which have been widely used for complex NLP tasks?

**Reasons To Accept:**

- This work achieved results surpass those previously reported in the literature for this task.

- Detailed analyses present original and interesting outcomes of argumentation processing. The realization that "learning representations from graph-structured data is a better idea than just using the whole text" holds significant importance within the current context of NLP research, wherein generative LLMs are garnering widespread attention.

- The incorporation of error analysis enriched the discussions.

- The text provides a concise and well-structured literature review on automatic debate processing.



**Reasons To Reject:**

- Despite the strengths of the proposed work, the attained F1 score by the best model still appears relatively low. As a matter of fact, this is not a poor outcome when considering the conventional models employed for the task, but it's possible that the proposed model might be outperformed by newer Generative LLMs. It would be important to discuss the influence of these novel models on the task's execution.

**Reproducibility:**

3: Could reproduce the results with some difficulty. The settings of parameters are underspecified or subjectively determined; the training/evaluation data are not widely available.

**Reviewer Confidence:**

4: Quite sure. I tried to check the important points carefully. It's unlikely, though conceivable, that I missed something that should affect my ratings.

**Typos Grammar Style And Presentation Improvements:**

- Line 452: has been carried OUR using Pandas

- Line 624-5: learning representations from A graph-structured data

---

> ### Author Rebuttal · Authors · 2023-08-24
>
> Dear Reviewer 5zCQ,
>
> First of all, we would like to thank you for your review and your constructive comments.
>
> In our response, we would like to address the two questions and the mentioned reasons to reject with hopes of improving your impression in our work.
>
> Regarding the first question, we considered the use of Support Vector Classifiers and Linear Regressors in our experiments. However, in our task setup, each debate represents a sample with an assigned label to predict the winning team (i.e., In favour or Against). Each debate has an average length of 4819 words (approximately 40 minutes of continued argumentative natural language being uttered by both teams). Such a complex input is a huge constraint for almost every NLP algorithm from classical Machine Learning to recent Deep Learning and generative LLMs. This is the main reason why it was not feasible to include simpler probabilistic models in our results section.
>
> The answer to the second question and the provided reasons to reject is closely related to the answer provided above. Our answer is that yes, we tried to use GPT-3.5-turbo in the automatic evaluation of debates, but the debates were longer than what the OpenAI API allowed us to use as a prompt. Therefore we were not able to test these approaches in our work. However, I am sure that it is going to be an interesting line of research to be explored in the near future.
>
> Thank you very much again for your time and dedication,
>
> The authors.

---

### Official Review · Reviewer_cvvq · 2023-08-05

**Soundness:** 3

**Excitement:**

3: Ambivalent: It has merits (e.g., it reports state-of-the-art results, the idea is nice), but there are key weaknesses (e.g., it describes incremental work), and it can significantly benefit from another round of revision. However, I won't object to accepting it if my co-reviewers champion it.

**Missing References:**

- [Graph Embeddings for Argumentation Quality Assessment](https://aclanthology.org/2022.findings-emnlp.306) (Marro et al., Findings 2022)

- [Yes, we can! Mining Arguments in 50 Years of US Presidential Campaign Debates](https://aclanthology.org/P19-1463) (Haddadan et al., ACL 2019)

**Paper Topic And Main Contributions:**

This paper proposes a new method for automatically evaluating and predicting the winning stance in professional argumentative debates. The key problem it addresses is performing debate analysis and evaluation on complex natural language arguments, which have been relatively unexplored.

The main contributions are:

- Introduces a hybrid technique combining concepts from argumentation theory and NLP to model complete reasoning lines in debates and determine a winning stance.

- Demonstrates superior performance of the hybrid technique over pure NLP or argumentation theory baselines on the debate evaluation task.

- Provides detailed experimental analysis and error analysis lending insight into the benefits of combining logical and linguistic knowledge for this problem.

Overall, this engineering-focused paper makes contributions in methods, and analysis towards advancing NLP for a complex argumentation problem.

**Questions For The Authors:**

A) Your proposed approach is evaluated only on a single dataset of Catalan debates. Can you discuss how you expect the hybrid technique to transfer to other debate domains and languages? Do you plan to test generalization capability on other debate datasets?

**Reasons To Accept:**

- Tackles a complex and underexplored NLP problem - evaluation of professional debates with complete logical argumentation - using a hybrid approach.

- The details of the hybrid technique combining concepts from argumentation theory seem sound and applicable to debates.

- Well-written with a clear explanation of all concepts for audiences unfamiliar with argumentation theory.

- Provides compelling evidence that bridging logical and neural techniques is beneficial for argument-mining problems.

- Could inspire more creative hybrid modelling and use of structured knowledge in NLP systems.

**Reasons To Reject:**

My biggest issue with this paper is the lack of comparison to other techniques for the debate evaluation task, which makes it difficult to situate the novelty of the proposed approach. The authors only compare against baselines they define, without discussing prior proposals from the literature. Given the lack of comparisons, it is unclear whether this approach substantially advances upon the state-of-the-art for this task.

Additionally, the literature review overlooks highly relevant prior work. The authors claim in line 110 that "Most of this research has been focused on performing an individual evaluation of arguments or argumentative lines of reasoning (Wachsmuth et al., 2017) instead of a global, interactive viewpoint where complete debates consisting of multiple, conflicting lines of reasoning are analysed". However, Wachsmuth et al. (2017) proposed a Reasonableness dimension that evaluates the whole argument, considering counter-arguments and rebuttals. In a debate scenario, counter-arguments and rebuttals are what constitute the bare bones of it so one would imagine that a dimension like Reasonableness would at least be discussed. Furthermore, Marro et al. (2022) also introduced a technique to assess the reasonableness of full argument graphs, combining text and graph structure. Yet neither of these directly related works are acknowledged or compared to. The lack of citations and comparisons to existing techniques that also evaluate full argument graphs raises concerns. The paper does not sufficiently situate its contributions among prior methods or demonstrate advancement over closely related approaches. A more rigorous review and comparison to previous debate evaluation and reasonableness assessment research is needed.

**Reproducibility:**

4: Could mostly reproduce the results, but there may be some variation because of sample variance or minor variations in their interpretation of the protocol or method.

**Reviewer Confidence:**

5: Positive that my evaluation is correct. I read the paper very carefully and I am very familiar with related work.

---

> ### Author Rebuttal · Authors · 2023-08-24
>
> Dear reviewer cvvq,
>
> First of all we would like to thank you for your insightful feedback and constructive comments. In our response, we would like to address the two mentioned reasons to reject and the question with hopes of improving your impression in our work.
>
> We would like to start my response by addressing your concern about the lack of some state-of-the-art related research by mentioning that we were aware of the mentioned works but we finally did not include them in our Related Work section because of the space limitation and their limited connection to our proposal. We wanted to make it clear since the very beginning of the paper the huge difference between the work done on the evaluation of individual arguments (even though they might contain some supporting/conflicting structures such as the ones included in Wachsmuth at al., 2017) from evaluating a complete argumentative debate where many different arguments are used by the teams representing two stances in a very interactive dialogue. Said that, we are willing to improve our work by including not only the mentioned references but other references such as (Lenz et al., 2019), (Kuhlmann and Thimm, 2019), (Malmqvist et al., 2020) and (Goffredo et al., 2023) where the concepts of argument mining, argumentation semantics, and argument graph networks are brought together. Given that in the camera ready version of the work we will have an additional page, the space limitation should not be a problem, and we will be able to improve the contextualisation of our work with other recent research in this direction.
>
> Another important aspect criticised in your review is the lack of direct comparison between our proposal and other techniques for the task of debate evaluation. This is a more difficult challenge, mainly due to the nature of existing annotated corpora containing argumentative debate annotations. As discussed in the Section 3 of our paper, the VivesDebate is the first and only publicly available argumentation corpus that contains the annotations for the entire debate, i.e. it annotates all Argument Discourse Units of the complete debate, thus being able to establish relationships along the complete lines of argumentative reasoning presented by the debaters, including in addition the objective scores of a professional jury on the argumentation presented in each debate by each team. This is the main reason why a direct comparison of our method with previous research has not been possible.
>
> So far, the existing debate datasets that contain relevant annotations to evaluate our method fall into two main categories: (i) contain short text-based debates/arguments (see Wachsmuth at al., 2017, Durmus and Cardie, 2019), or (ii) divide the argumentative debate transcripts into smaller snippets and note the argumentative structures and their relations in these snippets, but miss possible relations between arguments belonging to different text snippets, e.g. that have been proposed further apart in time (see Visser et al., 2020 or Hautli-Janisz et al., 2022).
>
> In shorter (and simpler) argumentations, fitted LLMs might be sufficient to provide convincing results. However, in full argumentative debates such as those included in the VivesDebate corpus, the use of such approaches is limited by the size of the natural language inputs. For this reason, we have not been able to directly compare the performance of our proposed method to any previous work. However, in our evaluation we have not limited ourselves to naive baselines (e.g. random or majority), but have included algorithms used in similar recent research, such as the Longformer.
>
> Finally, we would like to respond your question. The language and the domain can undermine the generalisation of the trained models included in our system. However, as long as corpus of complete debates with their corresponding evaluations are shared in English, a more complete system where the complete argument mining pipeline combined with our proposal can be trained and evaluated. However, given the nature of our proposal, it might not be informative to evaluate our proposal in other kinds of shorter/simplified argumentative debates such as the one released in (Wachsmuth at al., 2017) or (Durmus and Cardie, 2019), since a LLM will be able to provide better results. The most interesting part of our proposal is approaching the challenge that represents the size of complete long argumentative debates, and the way in which we encode these debates so that they can be used to train Neural Network-based algorithms (a graph network in our case). We are also very interested of evaluating the generalisation capability of our system on other languages/domains and in including spoken features into the evaluation algorithm in future works.
>
> We hope that with this response we will be able to improve your perception on the Soundness and Excitement of our work.
>
> Thank you very much again for your time and dedication,
>
> The authors.
>
> NEW REFERENCES
>
> Lenz, M., Ollinger, S., Sahitaj, P., & Bergmann, R. (2019). Semantic textual similarity measures for case-based retrieval of argument graphs. In Case-Based Reasoning Research and Development: 27th International Conference, ICCBR 2019, Otzenhausen, Germany, September 8–12, 2019, Proceedings 27 (pp. 219-234). Springer International Publishing.
>
> Kuhlmann, I., & Thimm, M. (2019, December). Using graph convolutional networks for approximate reasoning with abstract argumentation frameworks: A feasibility study. In International Conference on Scalable Uncertainty Management (pp. 24-37). Cham: Springer International Publishing.
>
> Malmqvist, L., Yuan, T., Nightingale, P., & Manandhar, S. (2020). Determining the Acceptability of Abstract Arguments with Graph Convolutional Networks. In SAFA@ COMMA (pp. 47-56).
>
> Hautli-Janisz, A., Kikteva, Z., Siskou, W., Gorska, K., Becker, R., & Reed, C. (2022, June). Qt30: A corpus of argument and conflict in broadcast debate. In Proceedings of the 13th Language Resources and Evaluation Conference (pp. 3291-3300). European Language Resources Association (ELRA).
>
> Goffredo, P., Cabrio, E., Villata, S., Haddadan, S., & Sanchez, J. T. (2023, June). DISPUTool 2.0: A Modular Architecture for Multi-Layer Argumentative Analysis of Political Debates. In Proceedings of the AAAI Conference on Artificial Intelligence (Vol. 37, No. 13, pp. 16431-16433).

---

### Meta-Review · Area_Chair_tYiF · 2023-09-18

**Recommendation:** 4

**Metareview:**

This paper proposes a new method for automatically evaluating and predicting the winning stance in professional argumentative debates. The key problem it addresses is performing debate analysis and evaluation on complex natural language arguments, which have been relatively unexplored.

The reviewers agree that the [hybrid] approach is very interesting and the results are strong. In addition, the paper was well-written and clear, with a generally strong literature review. The authors provide a clear error analysis which helps support the results and conclusions nicely. The hybrid approach is also technically sound and might provide inspiration for additional NLP tasks.

---

### Decision · Program_Chairs · 2023-10-07

**Decision:**

Accept-Main

**Comment:**

This paper proposes a new method for automatically evaluating and predicting the winning stance in professional argumentative debates. The key problem it addresses is performing debate analysis and evaluation on complex natural language arguments, which have been relatively unexplored.

The reviewers agree that the [hybrid] approach is very interesting and the results are strong. In addition, the paper was well-written and clear, with a generally strong literature review. The authors provide a clear error analysis which helps support the results and conclusions nicely. The hybrid approach is also technically sound and might provide inspiration for additional NLP tasks.